# The Experience of Patients with Type 1 Diabetes Mellitus with the Use of Glucose Monitoring Systems: A Qualitative Study

**DOI:** 10.3390/nursrep15080294

**Published:** 2025-08-12

**Authors:** Anxela Soto-Rodriguez, Ana Fernández-Conde, Raquel Leirós-Rodríguez, Álvaro Toubes Opazo, Nuria Martinez-Blanco

**Affiliations:** 1Family and Community Nursing, CS A Cuña, C/Sainza 42, 32005 Ourense, Spain; nuria.martinez.blanco@sergas.es; 2University Hospital Complex of Ourense, C/Ramon Puga Noguerol, 54, 32005 Ourense, Spain; ana.fernandez.conde@sergas.es (A.F.-C.); atoubesopazo2000@gmail.com (Á.T.O.); 3SALBIS Research Group, Nursing and Physical Therapy Department, University of Leon, 24400 Ponferada, Spain; rleir@unileon.es

**Keywords:** glucose metabolism disorders, blood glucose self-monitoring, quality of life, qualitative research, nursing

## Abstract

**Aim:** The purpose of this study was to explore the broad experience of continuous glucose monitoring from the perspective of patients diagnosed with type 1 diabetes mellitus, including not only their emotions and feelings but also the lifestyle changes, perceptions, and social aspects associated with its use. **Design:** This is a phenomenological qualitative study. **Patient or Public Contribution:** The sample consisted of 10 adult patients diagnosed with type 1 diabetes who had been using the continuous glucose monitoring system for at least 6 months and were patients of the Endocrinology and Nutrition Service of the University Hospital Complex of Ourense. **Methods:** The recorded interviews were conducted in November 2024. The conversations were audio-recorded with the participants’ consent, and then transcribed for thematic analysis. **Results:** Three main categories were identified: “experience prior to continuous glucose monitoring” (accessibility, prior knowledge, and expectations), “experience with the use of continuous glucose monitoring” (perception of healthcare support, concerns, strengths, and alarm management), and “experience regarding the disease” (self-management of the disease and safety). Despite the fact that diabetes mellitus is a complex chronic disease, all participants provided a positive assessment of their progress and improved control through continuous glucose monitoring. **Conclusions:** All participants felt more secure and protected with continuous glucose monitoring, improving their quality of life. The main concern among the subjects was the possibility of the sensor failing. They positively valued the alarm system in case of hypoglycemia. The CGM is a highly effective tool for the management and self-control of diabetes and promotes the relationship between patients and professional health. **Impact:** The findings of this study have important implications for clinical care, highlighting the need for more training and more health education at the first level of health care, such as health centers.

## 1. Introduction

According to the International Diabetes Federation Atlas, there are approximately 1.2 million people with type 1 diabetes mellitus (T1DM). This represents a significant increase in the number of cases of this disease in recent years [1]. It is essential for patients with T1DM to maintain adequate glycemic control, both to prevent acute complications, such as hypoglycemia or hyperglycemia, and to reduce the risk of chronic complications, including both microvascular and macrovascular issues [1,2].

At the same time, many people with diabetes have glycated hemoglobin (HbA1c) levels higher than those recommended for disease management and find it challenging to improve their glycemic control [1,2].

Continuous glucose monitoring (CGM) and fingerstick capillary glucometry are two widely used methods for glycemic control in patients with diabetes, each with specific characteristics and applications. On the one hand, fingerstick capillary glucometry is the traditional method for spot glucose measurement. Although invasive and repetitive, it offers high accuracy and low cost, with widespread availability. However, it does not provide continuous data or glycemic trends, which limits the early detection of variations and precise treatment adjustments [3,4]. In recent years, CGM devices have transformed T1DM management [2]. The CGM devices measure glucose levels in the interstitial fluid in real time. By having continuous information about glucose levels, people with diabetes can better manage their condition, achieving more effective disease control. These devices also offer the possibility of setting alerts and alarms that can help prevent or address acute complications such as hypoglycemia and/or hyperglycemia early, situations that can have serious consequences if not treated promptly [3].

From the patient’s perspective, monitoring provides short-term information that facilitates decision-making, as well as long-term information that allows for tracking disease progression, identifying patterns, and consequently better adjusting treatment [4,5]. Furthermore, for healthcare professionals, glucose monitoring offers additional information that helps evaluate treatment effectiveness, make adjustments, and provide personalized recommendations [6].

Today, we know that collaboration with healthcare professionals is essential for achieving optimal diabetes management with CGM devices. Educational programs are available to help people with diabetes interpret the data provided by the device and make the necessary adjustments to their diabetes management plan [4,6,7]. However, the emotional impact that using these devices can have on patients is often overlooked. Therefore, the aim of this study was to explore the broad experience of CGM from the perspective of patients diagnosed with T1DM, including not only their emotions and feelings but also the lifestyle changes, perceptions, and social aspects associated with its use.

## 2. Materials and Methods

A qualitative study with a phenomenological approach was conducted in accordance with the World Medical Association’s Helsinki Declaration for Human Studies (rev. 2024). The theoretical-methodological framework was inspired by Paul Ricoeur [8]. This approach was chosen because phenomenology focuses on describing and interpreting human experiences, opinions, beliefs, emotions, and values to understand the fundamental nature of those experiences. The data were handled anonymously and confidentially, and all participants in this study were informed of its objectives and provided written informed consent for their participation. This study was approved by the Ethics Committee for Research of Pontevedra-Vigo-Ourense (Registration Code: 2024/429).

### 2.1. Sample

The inclusion criteria were as follows: (a) adult patients diagnosed with type 1 diabetes mellitus (T1DM) aged over 18 years; (b) who had been under follow-up in the Endocrinology and Nutrition Service of the Hospital Complex of Ourense; and (c) had been using CGM for at least six months. Additionally, patients were excluded for the following reasons: (a) they had any illness or psychological condition that could prevent participation for health reasons; or (b) if they were unable to express themselves due to communication difficulties.

Patients who met the inclusion criteria were recruited and invited to participate in this study. Ultimately, a total of 10 participants were selected through purposive convenience sampling, which is commonly used in exploratory qualitative studies. While this type of sampling is appropriate for accessing individuals with direct and relevant experience of the phenomenon under investigation, it may limit the transferability of the findings, as the experiences collected are tied to a specific clinical context and a small group of participants. Therefore, the results should be interpreted in light of the particular characteristics of the setting in which they were obtained.

### 2.2. Procedure

Data collection was carried out through individual semi-structured interviews. Personal interviews were chosen due to the greater opportunity for privacy they provide compared to other techniques. The interviews were conducted face-to-face to establish a relationship between the researcher and the patient and to enhance data collection.

The interviewers used a guide based on relevant literature, which included some key questions to be addressed throughout the interview (Table 1). However, the order in which the various topics were discussed and the way questions were phrased were left to the discretion and judgment of the lead interviewer. The questions aimed to understand the emotions and feelings experienced by patients with T1DM who use CGM.

Data collection was always carried out by two interviewers simultaneously, with the aim of having multiple observers and being able to cross-check the collected data. Both interviewers are nursing professionals affiliated with the Endocrinology and Nutrition Service of the Hospital Complex of Ourense, one of whom has extensive experience in research and clinical interviewing and took on the role of principal investigator. The interviewers had no prior relationship with the participant.

To facilitate communication and establish a trustful environment, at the beginning of each session, the principal investigator explained the purpose and intentions of this study, provided the Study Information Sheet, and requested informed consent. Anonymity and data confidentiality were ensured by assigning each participant an anonymous identification code with a sequential number. Participants were also offered the possibility to withdraw from the session at any time without giving any explanation. Each interview lasted approximately 20 to 25 min. The conversations were recorded using a digital voice recorder to later facilitate the transcription of the dialogues. Once the transcriptions were completed, the recordings were destroyed.

Data collection was concluded when thematic saturation was reached, that is, when no new themes or relevant perspectives emerged in three consecutive interviews. This criterion is commonly accepted in qualitative studies and guarantees that a sufficient understanding of the phenomenon under investigation has been obtained. In our case, saturation was reached with the participation of 10 subjects, indicating that responses were beginning to repeat themselves and the codes and categories were consolidated. This process was monitored through ongoing inductive analysis throughout the interviews. The recorded interviews were conducted in November 2024 by the researchers at the Diabetes Therapeutic Education Consultation of the Hospital Complex of Ourense.

### 2.3. Data Analysis

The qualitative analysis followed the COREQ criteria. The data were transcribed verbatim by one of the interviewers within 24 h following each interview. To ensure the validity and reliability of this study, and to minimize potential biases stemming from the researchers’ perspectives on the topic, two researchers independently coded, summarized, and refined the interview material. This process followed Colaizzi’s [9] seven-step phenomenological method, through which primary and secondary themes were identified and organized. Discrepancies that arose during coding were discussed in meetings with experienced colleagues from the research team until consensus was reached. This approach helped strengthen the credibility, reliability, and confirmability of the analysis. Although no specific qualitative analysis software was used, a detailed and systematic record of all analytical decisions and reflections was maintained throughout the process, facilitating this study’s traceability and auditability. Additionally, field notes were incorporated to contextualize and enrich the interpretation of the data.

## 3. Results

A total of 10 participants were included in this study, and their main sociodemographic data are presented in Table 2. Among them, 60% were women, and the average age was 53.6 ± 8.4 years.

After analyzing the interviews, it was identified that the participants’ experiences and perceptions were related to one of the following themes: experience prior to using CGM, experience during the use of CGM, and experience related to the illness. Figure 1 presents the final summary of themes and subthemes, organized in a hierarchy defined by the level of abstraction. Below, the themes and subthemes are presented, accompanied by quotes from the participants.

(CGM: continuous glucose monitoring)Experience prior to CGM: this theme referred to the information previously received and the knowledge patients already had before their first contact with and use of this device. It also included the factors that, from their perspective, influenced their decision to start using it. In our study, we identified two subthemes:(a) Access to CGM: more than half (six participants) became aware of it through the Endocrinology service, although four participants stated that they learned about it through information from the internet or from other family members/friends who had used it previously:


*“… finally, they referred me to the endocrinologist, and the endocrinologist saw that it was diabetes. (…) Then, they sent me to diabetes education, and the first thing they gave me was the sensor.”*
(Participant 1)


*“… I had a friend who had it. Because I used to inject myself and I had many calluses from pricking myself so much every day, because I would prick myself 4 or 5 times a day. (…) A friend told me, let’s see if they can get it for you.”*
(Participant 4) [Note: resigned and tired tone.]

One of the patients mentioned that, even though they were not covered by the healthcare system years ago, they always tried to buy one despite its high price:


*“… it started to gain attention before the healthcare system included it. From time to time, I would buy one, because logically, my situation wasn’t that great.”*
(Participant 2)

(b) Prior knowledge and expectations: for many of the participants, the initial expectations regarding monitoring were that this procedure would promote greater control over their condition, and that they would finally be able to stop using the traditional method of capillary glucose monitoring through finger pricks:


*“As long as it meant no more pricks, anything would do.”*
(Participant 5)


*“… maybe it was easier. My fingers hurt a lot already.”*
(Participant 7)


*“If I’m being honest, I just thought, thank goodness I don’t have to prick my fingers.”*
(Participant 9) [Note: nods and smiles.]

On the other hand, one participant mentioned that at first, they found this innovative system a bit complicated and challenging:


*“Maybe it was going to be a bit of a hassle to use and it was going to be complicated.”*
(Participant 1) [Note: low voice, long pauses, downward gaze.]

Experience with the use of CGM: this theme grouped the sensations, perceptions, and data reported by patients once they started using this device for monitoring, including not only the main concerns or fears but also the strengths observed. Additionally, this category contains the perception of the quality of care from the nurse’s perspective. In our study, we identified five subthemes:(a) Perception of healthcare support: this category included data involving feelings related to perceived support and the care received from primary and specialized healthcare staff in general, regarding the use and management of the CGM device. All participants rated positively the accessibility they have to the Endocrinology and Nutrition Service in case they have any questions, not only about this device but also regarding uncertainties about their disease treatment. Furthermore, they appreciated that both the endocrinologist and nurse from this unit anticipated events, constantly informing them of how the disease would progress, which helped ensure optimal treatment and management of the situation. Many participants valued the telephone contact with this unit.


*“Both the endocrinologist and the nurse made everything crystal clear for me. They took the time to explain the whole process, what was going to happen, and how I would feel. We were always one step ahead. Nothing ever took me by surprise.”*
(Participant 8)


*“… I have a phone number to call for Diabetes Education (…) There’s so much information to process that there are always details that slip through. But I always have that phone number.”*
(Participant 1)


*“… you have easy access to the endocrinologist, they attend to you right away.”*
(Participant 4) [Note: nods and smiles.]

On the other hand, all participants reported problems and deficits in primary care centers, as they have limitations regarding the management of sensors, and consider that, being the first line of healthcare, they should be more prepared. The participants indicated that primary care healthcare professionals should be trained and have experience with these systems, which are becoming more common in our society. Additionally, all participants agreed on the importance of primary care healthcare professionals having more knowledge in such a specific and complex area as nutrition in T1DM:


*“… In the health center, there’s no one trained. The general practitioner doesn’t understand, and neither do the nurses. Unless they’re specialized in endocrinology, they don’t understand anything.”*
 (Participant 2) [Note: shakes head slowly.]


*“In the health center, maybe they don’t have the same training and can’t tell you much. You might end up explaining more about the sensor to them than they explain to you…”*
 (Participant 3) [Note: shrugs.]

(b) Concerns related to the management of the device: all patients expressed that they felt uneasy about the fear of it failing or the fear of being without it:


*“One of the main concerns is the fear that it will fail, you’re afraid it won’t sound the alarm at night.”*
(Participant 2) [Note: shows general tension on the face.]


*“I’m afraid, sometimes when I’m wearing short sleeves, that it might come off or get caught on something.”*
(Participant 4)

One of the patients mentioned that they experienced dependency on the device and would check it constantly:


*“… I was a bit obsessed with my blood sugar and checked it a lot (…) at first, it was like, all the time, constantly. But you don’t want them to take it away from you. It’s a love-hate relationship.”*
(Participant 1)

(c) Concerns related to the social and emotional areas: Four women expressed feeling uncomfortable or discriminated against in certain situations in their daily lives. Notably, one of them associated wearing the device with not being selected for a job position because they were identified as a sick person:


*“… I don’t like people seeing it. Especially in a job interview. (…) These are problems. Who wants a problem?”*
(Participant 2) [Field note: the interviewee avoided eye contact and appeared restless.]

Other female participants indicated feeling upset about experiencing unpleasant situations in their social context and being forced to explain the continuous glucose monitoring:


*“There was a comment near me from a woman who noticed I was wearing it and made a comment like, ‘Oh, you have a disease, right?’.”*
(Participant 7) [Note: unsure tone, trembling hands.]


*“People would ask me in the summer, ‘What’s that you’re wearing?’.”*
(Participant 10)

Male participants conveyed that they did not feel observed by others and did not give much importance to it, normalizing the situation:


*“At first, I did care if people noticed me, especially at the beach. But now I don’t. Now, I don’t mind wearing three here or four.”*
(Participant 9)


*“I don’t mind if it’s seen. The only thing people say is, ‘What’s that?’ Those who don’t know ask, ‘What is that?’ Now you pay more attention and notice that more people are wearing it.”*
(Participant 8)

(d) Strengths: the interviewed patients highlighted as the main strength the fact that they were able to avoid capillary finger pricks:


*“I no longer had to prick my finger. The device is not difficult to use.”*
(Participant 1)


*“Not having to constantly prick myself to check my blood sugar.”*
(Participant 3)

They also valued having the CGM always with them, as it syncs with their phone, describing it as something that is convenient, hard to forget (unlike the traditional glucometer), easy to use, and promotes autonomy (because it does not require the help or assistance of a third person):


*“The old machine, sometimes I’ve forgotten it at home (…) I put it on myself, I got the hang of it, and I don’t find it difficult at all. I understood it well from the first day.”*
(Participant 5)


*“You don’t need third parties to put it on.”*
(Participant 8)

Particularly, one of them mentioned that thanks to this innovative system, they feel motivated and eager to participate in different activities:


*“With this, I am much better. I look much better. Because before, I didn’t feel like doing anything, I would sit and watch TV, I didn’t go out, and on the days when I could maybe go out, I wouldn’t because I couldn’t.”*
(Participant 9)

(e) Alarm management: all the patients acknowledged that the sensor alarms, which alert about high and low blood sugar levels, are a very positive resource for managing the disease:


*“They are a peace of mind for me, and they don’t stop until you turn them off.”*
(Participant 4)


*“You get a little bored with them, but they are absolutely necessary and great.”*
(Participant 6)


*“The alarms at night make me feel safer (…) You have a low during the day, and that’s fine, but a low at night, when you end up waking up, and just thinking you don’t know if you’ll wake up…”*
(Participant 2) [Note: shows soft smile.]

However, six participants show that they only have the hypoglycemia alert active:


*“When I have it high, I’m not using it. Because when it’s high, I inject insulin, and that’s it. I prefer to have it off because if not, it keeps sounding continuously. And I keep the low alarm on because I think it’s more serious.”*
(Participant 2)

Although some mentioned that these notifications cause them nervousness and anxiety and that at night they prefer to keep only the low glucose alarm, as they generally consider it absolutely necessary:


*“The low glucose alarm, I think it’s important to have it on.”*
(Participant 1)


*“It’s a way to notice that something is happening, although when it suddenly rings, it makes me anxious and I get more nervous.”*
(Participant 7)

Only one of them was resistant to all the alarms, even the one that alerts about hypoglycemia, because they felt they perceive and recognize the characteristic symptoms of this situation without relying on the sensor:


*“They are disconnected, the endocrinologist had told me that they could even make me a little nervous and that it wasn’t worth it. (…) For now, I recognize hypoglycemia right away. And at night, I prefer to sleep peacefully without the shock of the alarm.”*
(Participant 10)

Experience regarding the disease: This includes the sensations and reactions triggered in the patient with CGM and how this device influences the course of their T1DM. The results ranged from empowerment and self-management of the disease to the sense of security in daily life provided by the device:(a) Empowerment tool for self-management of the disease: Once the disease is accepted, the patient must adjust to this new situation, a new and unknown scenario for many that they have to adapt to, and which initially seems complicated (as conveyed in other categories as well). All the patients with T1DM in this study considered the CGM as a relief, which reduced their concern and helped them face the disease more effectively. All of this, even though for many of them it was a new reality they had to learn to live with, assuming its potential complications and taking an active role in managing the new condition:


*“Having control over glucose, knowing how you are at any moment. If you’re going to exercise, then I have to take something, or I don’t have to take it…”*
(Participant 10) [Note: nods and smiles.]

All the subjects made a positive assessment of their progress over time and felt strong enough to achieve a good quality of life despite their chronic illness:


*“The comfort of being able to improve my diabetes, to improve the graph, improve my quality of life.”*
(Participant 2)


*“… It makes life so much easier for the user. It helps with better control of diabetes. (…) It’s much easier and more continuous.”*
(Participant 6)


*“… This was a lifesaver… I never thought it would be so useful and valuable to me. For me, it’s the best thing that could have happened.”*
(Participant 9)

All of them indicated that this technology favored greater self-control of the disease and improved their blood sugar levels, making their daily life easier. They considered the CGM a unique strategy for managing out-of-range glucose levels, applying their knowledge to achieve optimal blood sugar values again:


*“If I see that it’s rising, then I correct it. I mean, it’s telling me what I need to do all the time.”*
(Participant 1) [Note: with a confident tone.]


*“It makes my day-to-day life easier. Every so often, I check the app to see if I’m doing well. (…) It’s a big step forward because it helps me a lot.”*
(Participant 8)

(b) Safety: the participants stated that the constant information provided by this device gives them great security, and they feel calmer regarding the risk of experiencing hypoglycemia:


*“I’m more insecure without the sensor.”*
(Participant 1)


*“At night, if you have hypoglycemia and you’re using another system, maybe you keep sleeping (…) with the sensor, you wake up or you wake up.”*
(Participant 6) [Note: with a confident tone.]


*“You relax, so to speak, because you know you have it, you know if you’re going to have a hypo or a hyper. (…) I think you live more peacefully.”*
(Participant 8)

The patients, regardless of their age, were aware that they were being monitored and felt more secure and protected:


*“I’m more calm, I know it’s there, I feel safe.”*
(Participant 7)


*“It’s not the first time that (…) it alerted me at 4 or 5 in the morning that I was low, and otherwise, I wouldn’t have noticed.”*
(Participant 4)

The greatest fear they revealed was the risk of hypoglycemia at night, which is why the vast majority highlighted that the main benefit of the sensor is that it alerts them if they experience nocturnal hypoglycemia. This is what gives them the greatest sense of security with this device.


*“You have a low during the day, and that’s fine, but a low at night, when you end up waking up, and just thinking you don’t know if you’ll wake up…”*
(Participant 2)

## 4. Discussion

This study investigated the experience and feelings of patients with type 1 diabetes who use CGM. Previous studies mainly explored patients’ opinions on the utility and satisfaction with these sensors using quantitative measures [7,10]. However, qualitative studies allow for a deeper understanding of how this device can individually affect the lives and emotions of people with T1DM. Although there are some studies that employ this methodology, their number remains very limited [5,11].

Regarding the experience prior to CGM, it should be taken into account that the use of CGM through an electronic device is a very recent advancement that helps manage T1DM [10,12]. Furthermore, as observed in this research, awareness of it remains limited. For this reason, it would be necessary to implement informational sessions, even for healthcare professionals, to increase literacy about these devices. Vloemans et al. [11] mention that familiarization with the device is part of the adaptation process, but the training received is not evaluated. Professional support and technology education are key to improving the user experience [5]. It has been demonstrated that keeping glucose levels as close as possible to the normal range delays the onset and slows the progression of T1DM complications such as retinopathy, kidney disease, neuropathy, and heart disease [3,4]. Therefore, the proper and supported use of technologies such as CGM is essential to optimize clinical outcomes and improve patients’ quality of life. Additionally, continuous access to real-time data allows for the identification of changes and patterns that previously went unnoticed, thereby facilitating better self-management.

What was identified in the subtheme prior knowledge and expectations reveals suboptimal adherence to traditional capillary blood glucose monitoring, leading to poor disease control, consistent with previous research [5,13,14]. This lack of compliance may be due to multiple factors, including the discomfort and inconvenience associated with frequent punctures, as well as the difficulty in integrating this practice into patients’ daily routines. In the work of Overend et al. [15], users highlighted the greater comfort and convenience of the system compared to traditional punctures for glucose measurement. Most participants described CGM as a simple and reliable method from the outset, providing comfort, peace of mind, and being essential for achieving appropriate glucose ranges and avoiding hypoglycemia. Therefore, this new method could improve treatment adherence in patients with T1DM due to its ease of use. This sense of security is especially valuable in avoiding episodes of hypoglycemia, which can be dangerous and difficult to detect without continuous monitoring. The work of Vloemans et al. [11], although it does not directly measure adherence, suggests that a better understanding of glucose levels and early detection contribute to more effective self-care. Additionally, participants report that the CGM provides them with greater safety and confidence, especially those with impaired hypoglycemia awareness, since they face a greater risk by not recognizing the symptoms of low glucose levels. Training not only improves the technical handling of the device but also strengthens patient confidence, reduces disease-related anxiety, and encourages active participation in daily self-care.

Regarding the experience using CGM, patient opinions reflect the importance of effective communication between healthcare professionals and patients. This communication should be facilitated through various channels to prevent users from experiencing feelings of abandonment or lack of attention, a problem identified in the current literature [14]. For this purpose, telephone follow-up for metabolic control in patients with T1DM has shown effectiveness similar to that of in-person follow-up [15,16,17,18]. Furthermore, it is important to tailor healthcare support and interventions for patients beginning to use CGM, adapting to the educational and pedagogical needs of diverse populations [12]. Cross-platform applications allow constant and immediate interaction and feedback with healthcare professionals, helping patients better understand glycemic variations and how to address them, thereby increasing their disease awareness [4]. Therefore, interaction with the medical team is essential for the correct interpretation of data and decision-making in the management of CGM. Effective communication is a fundamental pillar for optimizing the user’s experience with this technology. Clear communication not only helps resolve technical doubts but also reduces anxiety, facilitates learning, and promotes greater adherence to treatment [5,11]. In this study, many patients especially valued the telephone accessibility of the hospital’s endocrinology service.

Unlike other studies, this research explores the dissatisfaction or discomfort toward primary care health professionals, highlighting the perception that most nurses at this level lack knowledge about new technologies in diabetes care. The concerns revealed by patients support the need to inform and prepare them to know how to act if the sensor fails, encouraging them to interpret this as a positive experience, enhancing their sense of self-control and fostering active coping with the situation [13]. Additionally, diabetic patients tend to have a poorer perception of quality of life [19], which is associated with more depressive symptoms, lower treatment adherence, and worse disease management [20]. Specialized Family and Community Nursing professionals are ideally positioned to address the significant limitations described by the participants. Therefore, the role of nursing in Primary Care is essential to improve the quality of care for individuals, families, and communities in a close and continuous manner [18]. For this reason, it is crucial that these professionals are properly trained in new technologies for diabetes treatment [5,11].

This study, unlike previous research, focuses specifically on social challenges that hinder the adoption of technologies in diabetes treatment. Among the main social issues associated with the use of CGM are stigma, impact on body image, inequalities in access, lack of support from the social environment, and the digital or educational divide. These factors can affect the acceptance, proper use, and effectiveness of the device in diabetes management [5]. The concerns related to the social and emotional sphere identified suggest the need for a comprehensive approach that addresses not only physical needs but also psychological ones through a multidisciplinary framework. Furthermore, it is necessary to raise awareness of this disease and the devices it relies on (such as the CGM sensor) to normalize their use in society [20,21]. Previous studies have not conducted a detailed or specific analysis based on the sex or gender of participants regarding the use of CGM. It is worth noting that in this study, the only participants who expressed concerns in the social and emotional areas were women. Although the prevalence of T1DM is slightly lower in women compared to men, its clinical impact is significantly greater at every stage of life. This is because sex plays a role in managing cardiovascular risk factors, the progression of the disease, and the development of macrovascular and microvascular complications [22,23]. Healthcare professionals should consider these specificities to improve the quality of care for women.

Among the main strengths, it appears that CGM makes T1DM more manageable, reducing distress, as patients experience better disease control. This helps them feel more hopeful and positive. In fact, HbA1c levels [22] and carbohydrate consumption [24] seem to be lower in CGM users compared to those performing capillary punctures. Therefore, it is an empowering tool that allows patients to make informed decisions about their lifestyle, promoting engagement in self-care activities. By providing information about glycemic variability, CGM seems to help patients become more aware of the causes and effects of their glucose levels [1,2,22,23]. Furthermore, it has been shown that CGM reduces anxiety and stress in the patient’s family members by providing real-time information that enables safer and more proactive diabetes management, thereby improving the quality of life for everyone involved [24,25,26]. However, despite being considered an easy-to-use method, education and guided training would be necessary to reinforce and improve adherence to its use. This is particularly important for T1DM patients who are not accustomed to managing technologies, such as mobile applications.

On the other hand, patients valued the CGM alarms positively as a key tool for managing their diabetes, as they provide peace of mind and a sense of security, especially during the night, consistent with other studies. Some patients reported anxiety in response to the notifications and occasionally disable the hyperglycemia alarms to avoid discomfort from constant alerts. This behavior may limit the full benefit of the device and highlights the need for personalized strategies to optimize alarm settings according to each user’s preferences [5,11,15].

Regarding the limitations of this research, it should be noted that the sample did not include young patients, for whom body image alterations could be a significant factor. Adolescence is one of the most complex stages of life due to the numerous changes, not only physiological but also psychosocial, that occur [19,21]. Additionally, since only patients who voluntarily agreed to participate were involved, this could lead to a selection bias where these patients were more motivated, potentially influencing the types of responses obtained. On the other hand, we did not implement additional methods, such as participant verification of the results or other forms of triangulation (data or methodologies). Another limitation of this study was the lack of geographic and demographic diversity, which was attributable to methodological and logistical factors. Although theoretical saturation was reached, supporting the internal validity of the findings, we acknowledge that a broader and more heterogeneous sample could have enriched the understanding of the phenomenon under study. Therefore, future research should consider including participants with greater variability in these aspects to enhance the generalizability and applicability of the results.

However, as a strength of our study, it is worth highlighting that, while there is previous research on the experience of CGM in patients with T1DM, this is the first qualitative study to explore this experience from a comprehensive perspective and in a specific context (a Spanish city), conducting in-person, individual interviews. These methodological aspects allow for new findings and relevant nuances that complement and expand current knowledge on the use of CGM in this population.

## 5. Conclusions

Patients who use CGM feel more secure and protected. Additionally, they expressed feeling calm and relieved in managing their disease thanks to CGM. The greatest discomfort was that, although all participants positively assessed the accessibility to the Endocrinology and Nutrition Service, they felt there was a lack of support in their Primary Care centers in case they had any questions regarding this device or the treatment of their condition. The CGM is a very effective tool for the management and self-control of diabetes and promotes the relationship between patients and healthcare professionals.

## Figures and Tables

**Figure 1 nursrep-15-00294-f001:**
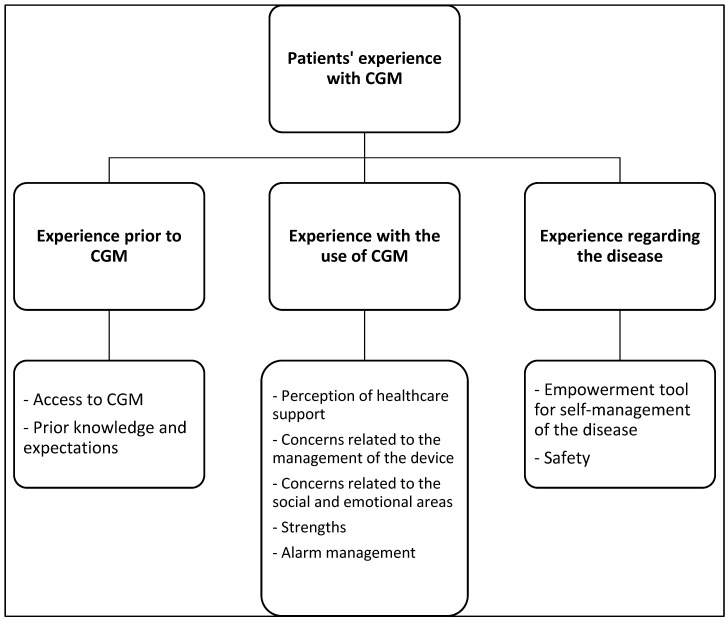
Outline of themes, subthemes, and categories.

**Table 1 nursrep-15-00294-t001:** Interview guide.

Question	Objective
How did you first learn about the continuous glucose monitoring device?	Explore the origin of knowledge about the device and initial impressions.
Did you accept being a user of this device right away or did you have any doubts?	Analyze the initial reactions and possible barriers to accepting the device.
What were your expectations of this sensor?	Identify the user’s expectations prior to using the sensor.
Were those expectations met?	Assess whether the device met the user’s expectations.
What is the most positive aspect of this sensor?	Highlight the beneficial aspects of the sensor from the user’s perspective.
What is the most negative aspect of this sensor?	Identify any negative points or areas for improvement according to the user.
What concerns did you have about using this sensor?	Address any worries or concerns prior to using the sensor.
What was the easiest part of using this sensor?	Determine what aspects of use were intuitive or simple.
And what was the most difficult?	Explore the challenges or difficulties encountered while using the device.
Do you think the support received from healthcare professionals was sufficient?	Evaluate the quality and adequacy of the support provided by healthcare professionals.
What do you think of the sensor alarms?	Investigate the user’s perception of the alarms, their usefulness, and possible drawbacks.
Would you improve any aspect of the sensor?	Gather suggestions for improvements or innovations based on the user’s experience.

**Table 2 nursrep-15-00294-t002:** Characteristics of the participants.

Particpant	Sex	Age	Level of Studies	Employment Status	Marital Status	Lives Alone	Time Since Diagnosis	Time Wearing the Sensor
1	Woman	49	University	Active	Single	Yes	1 year	7 months
2	Woman	50	Primary	Active	Married	No	22 years	3 years
3	Woman	47	Primary	Active	Married	No	18 years	2 years
4	Man	69	Primary	Retired	Widowed	Yes	25 years	8 months
5	Woman	63	Primary	Retired	Married	No	30 years	3 years
6	Man	49	University	Pensioner	Single	Yes	20 years	2 years
7	Woman	56	Primary	Active	Divorced	No	5 years	2 years
8	Man	42	Secondary	Active	Single	No	7 months	7 months
9	Man	61	Primary	Retired	Married	No	35 years	1 year
10	Woman	50	University	Active	Married	No	1 year	6 months

## Data Availability

The data presented in this study are available on request from the corresponding author.

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
