# Peer review of "The Experience of Patients with Type 1 Diabetes Mellitus with the Use of Glucose Monitoring Systems: A Qualitative Study"

_nursrep, 2025, doi:10.3390/nursrep15080294_

Round 1
Reviewer 1 Report
Comments and Suggestions for Authors
Thank you for the opportunity to review this qualitative study exploring the experience of patients with T1DM using CGM. Since the use of CGM is increasing, the experiences of patients could provide new insights into this device. However, I have some concerns.
METHODS:
Please provide more details on how data saturation was assessed. Line 120 - did this mean that no new themes emerged in the last three interviews? Including a description of your saturation process would improve methodological clarity.
While purposive convenience sampling is acceptable, consider briefly discussing how this might affect the transferability of findings.
I am wondering whether any field notes or observations were recorded alongside the interview transcripts.
The use of Colaizzi’s method is appropriate, and it is good to see that coding was done independently by two researchers. However, more detail is needed about how themes were developed-was qualitative software used? If analysis was manual, please explain the process. It would strengthen the manuscript to explain how credibility, dependability, and confirmability were addressed.
To enhance transparency and completeness in reporting, I recommend that the authors submit a completed reporting guideline checklist for qualitative research depending on which best aligns with their methodological approach.
Additionally, regarding Lines 103–104, the authors mention that the interviews aimed to explore the emotions and feelings of patients with T1DM using CGM. However, based on the Results section, the identified themes appear to address a broader scope of experiences, such as lifestyle changes, perception, and social aspects, rather than focusing solely on emotional responses. I suggest the authors either (1) align the stated aim more clearly with the broader experiential focus reflected in the findings or (2) clarify how emotions and feelings were represented within each theme.
INTRODUCTION:
Lines 70–71, the authors mention that the emotional impact of CGM use in patients with T1DM has been overlooked in existing research. However, in the sentence that follows, the stated aim of the study is described more broadly as "to explore the experience of CGM use among patients with T1DM." This creates a slight mismatch between the identified research gap and the aim.
DISCUSSION:
Lines 395–397, the authors state that this is the first qualitative study exploring the experience of CGM among patients with T1DM. However, several qualitative studies on this topic have already been published, and at least one review of such studies is available in the literature.
I suggest the authors to review and integrate relevant previous qualitative studies on CGM use in T1DM populations. Clarify what specifically differentiates this study to emphasize its originality. Doing so will provide appropriate context and demonstrate the contribution of this study to the existing body of knowledge in a more robust way.
Vloemans, A. F., van Beers, C. A. J., de Wit, M., Cleijne, W., Rondags, S. M., Geelhoed-Duijvestijn, P. H., Kramer, M. H. H., Serné, E. H., & Snoek, F. J. (2017). Keeping safe. Continuous glucose monitoring (CGM) in persons with Type 1 diabetes and impaired awareness of hypoglycaemia: a qualitative study. Diabetic medicine : a journal of the British Diabetic Association, 34(10), 1470–1476. https://doi.org/10.1111/dme.13429
Natale, P., Chen, S., Chow, C. K., Cheung, N. W., Martinez-Martin, D., Caillaud, C., Scholes-Robertson, N., Kelly, A., Craig, J. C., Strippoli, G., & Jaure, A. (2023). Patient experiences of continuous glucose monitoring and sensor-augmented insulin pump therapy for diabetes: A systematic review of qualitative studies. Journal of diabetes, 15(12), 1048–1069. https://doi.org/10.1111/1753-0407.13454
Author Response
The response to Reviewer 1 is included in the attached pdf document

Reviewer 2 Report
Comments and Suggestions for Authors
Current study “The experience of patients with type I diabetes mellitus with the use of glucose monitoring systems: a qualitative study”. contributes valuable knowledge to the field by capturing the lived experiences and perceptions of patients with type I diabetes about continuous glucose monitoring system. Here are few comments:-
- Abstract: line 35 here is typo error “MCG” it should be “CGM” (Continuous Glucose Monitoring).
- Line 35 “tool very effective” should be highly effective tool
- Introduction: CGM should also be compared with finger-prick tests
- Number of participants is very less. Adult patients should also be included in current study. A larger and more diverse sample could provide broader insights
- In inclusion criteria age of patients is missing
- the lack of geographic and demographic diversity (e.g., age, socioeconomic status, duration of disease) should also be included
Author Response
the response to reviewer 2 is included in the attached pdf document

Round 2
Reviewer 1 Report
Comments and Suggestions for Authors
All concerns were addressed. Thanks for revising the manuscript based on the comments.